# In Silico Analysis of Genes Associated with the Pathogenesis of Odontogenic Keratocyst

**DOI:** 10.3390/ijms25042379

**Published:** 2024-02-17

**Authors:** Carla Monserrat Ramírez-Martínez, Itzel Legorreta-Villegas, Claudia Patricia Mejía-Velázquez, Javier Portilla-Robertson, Luis Alberto Gaitán-Cepeda, Jessica Tamara Paramo-Sánchez, Osmar Alejandro Chanes-Cuevas, Alejandro Alonso-Moctezuma, Luis Fernando Jacinto-Alemán

**Affiliations:** 1Oral Medicine and Pathology Department, Postgraduate and Research Division, Dentistry School, National Autonomous University of Mexico, Mexico City 04510, Mexico; c.d.carlarmzmtz@hotmail.es (C.M.R.-M.); itzel.legovi@gmail.com (I.L.-V.); dramejiavelazquez@outlook.es (C.P.M.-V.); jpr@unam.mx (J.P.-R.); lgaitan@unam.mx (L.A.G.-C.); jessytam_1118@hotmail.com (J.T.P.-S.); 2Dental Biomaterials Laboratory, Postgraduate Division, Dental School, National Autonomous University of Mexico, Mexico City 04510, Mexico; oachanesc@gmail.com; 3Oral and Maxillofacial Surgery Department, Postgraduate Division, Dental School, National Autonomous University of Mexico, Mexico City 04510, Mexico

**Keywords:** odontogenic keratocyst, bioinformatics, extracellular matrix, viral infection

## Abstract

Odontogenic keratocyst (OK) is a benign intraosseous cystic lesion characterized by a parakeratinized stratified squamous epithelial lining with palisade basal cells. It represents 10–12% of odontogenic cysts. The changes in its classification as a tumor or cyst have increased interest in its pathogenesis. Objective: Identify key genes in the pathogenesis of sporadic OK through in silico analysis. Materials and methods: The GSE38494 technical sheet on OK was analyzed using GEOR2. Their functional and canonical signaling pathways were enriched in the NIH-DAVID bioinformatic platform. The protein–protein interaction network was constructed by STRING and analyzed with Cytoscape-MCODE software v 3.8.2 (score > 4). Post-enrichment analysis was performed by Cytoscape-ClueGO. Results: A total of 768 differentially expressed genes (DEG) with a fold change (FC) greater than 2 and 469 DEG with an FC less than 2 were identified. In the post-enrichment analysis of upregulated genes, significance was observed in criteria related to the organization of the extracellular matrix, collagen fibers, and endodermal differentiation, while the downregulated genes were related to defensive response mechanisms against viruses and interferon-gamma activation. Conclusions. Our in silico analysis showed a significant relationship with mechanisms of extracellular matrix organization, interferon-gamma activation, and response to viral infections, which must be validated through molecular assays.

## 1. Introduction

The odontogenic keratocyst (OK) is defined as a cyst of odontogenic origin characterized by its parakeratinized stratified squamous epithelial lining with nuclear hyperchromatism in its basal cells organized in a palisade [1]. OK was first described in 1876 and named by Phillipsen in 1956, who associated it with a tendency to recur [2,3]. In the third edition of the WHO classification of head and neck tumors, published in 2005, OK changed its designation from developmental cyst to benign odontogenic neoplasm, replacing its name with keratocystic odontogenic tumor, based mainly on its aggressive behavior, high recurrence, and mutations in the PTCH gene [4]. However, in 2017, it was reclassified as an odontogenic cyst due to evidence showing that the PTCH gene mutation could be found in non-neoplastic lesions, including dentigerous cysts [5,6].

It has been reported that this odontogenic cyst is the third-most common cyst [1,7]. In Mexico, their epidemiological trend is the same, since various reports place it as one of the most frequent odontogenic cysts [8,9]. With respect to age, OK could occur at all ages, with a peak incidence in the second and fourth decade of life, with a slight male predilection [1,7]. Their etiopathogenesis is related to intraosseous epithelial dental lamina remnants [10]. The anatomical region most affected is the mandible, preferably in the angle and ramus, which represents approximately 10% of mandibular cysts [11]. Clinically, OK could present as asymptomatic with variable soft tissue swelling, with or without pain, displacement of teeth, asymmetry, and occasionally paresthesia of the lower lip. The cystic bony expansion could be progressive, with very minimal changes at the initial stage and a typical growth in the anterior–posterior direction of medullar bone [1,7].

Radiographically, it can be observed as a radiolucent, well-defined, unilocular or multilocular area, with well-corticalized margins [2]. Histologically, it is made up of a cystic lumen, which contains scaly/wavy keratin, with a lining of parakeratinized squamous epithelium, with a thickness of 5 to 10 cell layers. In the basal layer, cubic or columnar cells are present in a palisade with polarization of the nucleus [3,12]. Although OK can be considered as benign lesions, a significant increase in volume can be observed in the bone mass due to its indolent symptoms, and recurrence can occur frequently in 0 to 62% of cases [13]. OK can occur in patients with Gorlin–Goltz Syndrome or Nevoid Basal Cell Carcinoma Syndrome, in which basal cell carcinomas, macrocephaly, congenital malformation of the lip or cleft palate, and polydactyly, as well as the symmetrical presence and high recurrence of OK, are some of the syndromic features [1]. It has been reported that PTCH1 mutations could be present in sporadic and nevoid basal cell carcinoma syndrome-related OK, as well as alterations in methylation pattern in p21, a potent cyclin-dependent kinase inhibitor that encodes for a protein that binds to and inhibits the activity of cyclin-CDK2 or -CDK4 complexes; this epigenetic change could be partially responsible for the proliferative OK pattern [14,15].

The controversy surrounding its cystic or tumorous nature and the distinct progression of its sporadic and syndromic forms make OK an important and fascinating subject for a pathogenic investigation employing fresh analysis techniques. The present study aimed to identify crucial genes involved in the pathogenesis of sporadic OK through in silico analysis.

## 2. Results

A total of 768 differentially expressed genes (DEG) with an FC greater than 2 and 469 DEG with an FC less than 2 were identified (Figure 1).

### 2.1. GO and KEGG Enrichment

The enrichment analysis using GO of DEGs with FC > 2 showed significant criteria related to cell adhesion, organization of collagen fibers, and organization of the extracellular matrix, among others, while the significant criteria of the KEGG analysis were protein digestion and absorption, ECM–receptor interaction, and the PI3K-Akt signaling pathway (Table 1).

For the DEGs with FC < 2, significance was observed in GO criteria related to innate immune response, defense response to viruses, and inflammatory response, among others, while in KEGG analysis, the significant pathways were Influenza A, Staphylococcus aureus infection, and allograft rejection, among others (Table 2).

### 2.2. IPP Network and Modular Analysis

A total of 198 and 106 DEGs were identified in >2FC and <2FC, respectively, which were entered into the STRING platform for the construction of the IPP networks, executing the analysis using Cytoscape-MCODE of genes with a score > 4, obtaining 46 and 24 genes derived from the >2FC and <2FC cuts, respectively (Table 3).

### 2.3. Post-Functional Enrichment of Candidate Genes

The Cytoscape-MCODE-obtained genes were post-enrichment analyzed to obtain upregulated genes (*n* = 34) that indicated a central relationship with extracellular matrix organization processes, endodermal formation, and cellular response to amino acid stimuli. The most frequently upregulated genes observed in the biological process criteria were COL5A2, MMP2, COL11A1, COL12A1, COL5A1, COL4A2, COL8A1, COL1A1, COL1A2, COL3A1, and COL6A1 (Table 4).

Regarding the post-enrichment of downregulated genes (*n* = 23), a main relationship was observed between virus response and cellular response to interferon-gamma, among others. The most frequently downregulated genes observed in the biological process criteria were BST2, CCL5, TNF, IFIH1, MX1, OAS1, OAS2, RSAD2, and STAT1 (Table 5).

## 3. Discussion

Epidemiologically, OK presents high variation with respect to its prevalence; it is considered that it may occupy third place of all cysts that develop in the mandible and maxilla. Its age of involvement ranges from the second to the fourth decade of life, with greater involvement in the male gender [7]. It has been considered that OK etiology is related to dental lamina cellular remains located mainly in the periodontal ligament and basal sites of the oral epithelium, and that in these cellular remains, genetic changes can occur, particularly mutations that allow the overexpression or inhibition of genes that lead to OK development. Among the genes involved in its pathogenesis, p53, p63, PCNA, Ki-67, PTCH, SMO, MMP1, and MMP2 have been reported [1,7]. Specifically, the presence of mutations in p53 and p63 has been related to greater aggressiveness in OK [16,17]. The expression of PCNA and Ki67 has been related to a greater proliferative potential, which has been considered a characteristic that gives greater clinical aggressiveness [18,19,20]. The identification of mutations in PTCH and SMO, members of the SHH signaling pathway, can be considered an indicator of the presence of LOH (loss of heterozygosity). Anecdotally, these mutations were one of the reasons why OK was classified as an odontogenic tumor; however, observing its presence in other benign non-tumor lesions, including the dentigerous cyst, allows us to postulate these mutations with cystic growth [21,22]. Our enrichment analysis using GO and KEGG presented criteria related to signaling pathways important for cancer development, such as the Wnt signaling pathway, although criteria related to the modification of the extracellular matrix were predominantly observed. The molecules related to ECM modification, extracellular matrix metalloproteinases (MMPs), have been involved in the degradation and modification of collagen fibers, which could be considered as a tissular characteristic associated with cystic growth [23]. Our in silico analysis showed the upregulation of MMP2 and MMP13. This coincides with Wahlgren et al.’s report, which suggested that the above MMPs, together with laminin 5 expression in the basement membrane, are important for cystic development [24].

Laminins are extracellular matrix glycoproteins involved in a wide variety of biological processes, including cell adhesion, differentiation, signaling, neurite outgrowth, and metastasis. The possibility has been suggested that laminin present in odontogenic epithelial remains influences proliferative activity in addition to acting as a chemoattractant for stromal and vascular cells, thus modulating the interactions between the epithelium and mesenchyme that favor the growth of these epithelial remains [25].

Of the 34 upregulated genes derived from post-enrichment, 30 had a direct relationship with modulation of the ECM and participation in other biological processes such as cell adhesion and proliferation. Of the genes related to cell adhesion, it has been reported that fibronectin, a dimeric glycoprotein soluble in plasma that participates in the processes of cell adhesion and migration, has already been detected in OK, associating its presence with more aggressive biological behaviors of the cyst through promoting a proliferative environment [26,27].

In relation to collagen’s expression pattern, our analysis showed the expression of fibrillar (COL5A2, COL11A1, COL12A1, COL5A1, COL1A1, COL1A2, COL3A1, and COL6A1) and basement membranes (COL4A2 and COL8A1)-related proteins. Cota et al. analyzed the expression of collagen VII, a fibril restricted to the basement zone beneath stratified squamous epithelia; however, they found a discontinuous immunoexpression pattern that could be associated with the expression of other collagen subtypes and by the collagenolysis related to MMP2 and MMP13 [28]. It is possible that the potential of small satellite or daughter cysts development by the budding of the basal layer could be related to that affection of the basement zone; this is a histological feature and behavior that could favor OK recurrence. A differential expression and behavior of MMP in sporadic and syndromic OK have been reported, suggesting that nevoid basal cell carcinoma syndrome OK could express an increased pattern, which could result in more aggressive growth [29].

In addition, the underregulated genes analysis showed that BST2, a molecule involved in the growth and development of B-cells and one that can inhibit the cell surface proteolytic activity of MMP 14, is underregulated. Normally, IFN-α could induce BST-2 to decrease the MMP2 activity by binding the cellular C-terminus of MMP 14 and inhibiting its proteolytic activity; it is possible that underregulation of BST2 could favor the non-restricted MMP2 activity [30]. Considering the above results, it is possible that sporadic OK could present an MMP2- and BST2-specific profile associated with extracellular matrix modification.

CCL5 and TNF are cytokines related to chemotaxis and proinflammatory response, respectively. Their expression has been associated with the infiltration of inflammatory cells into radicular cysts [31,32]. Zhong et al., through an OK bioinformatic analysis, showed a lower abundance of immature dendritic cells, CD56dim natural killer cells, activated dendritic cells, type 17 T helper cells, and neutrophils, and a higher presence of CD56bright natural killer cells, memory B cells, and natural killer T cells [33]. The underregulation of both cytokines could be associated with differential cell infiltration; this feature may be responsible for the immune ecosystem modulation that could impact on osteoclastogenic effects [34].

Our post-enrichment analysis showed that OK could have susceptibility or altered molecular mechanisms against viruses. IFIH1, MX1, OAS1, OAS2, RSAD2, and STAT1 as underregulated genes could be responsible for deficient antiviral activity. This is the first time that these genes have been related to OK; however, the viral infection in this cyst has been reported. Alsaegh et al. reported a higher proportion of EBV and KSHV co-infection in OK compared to other odontogenic cystic and tumor lesions [35]. The presence of this virus in OK can be explained by the detection of chronic inflammatory infiltrate with B cells, and in the particular case of KSHV, the endothelial and epithelial cells could be the targets of infection [36,37]. Supposing that viral infection may be part of cystic pathogenesis is very risky; however, there is indirect evidence of viral infection. Such is the case reported by Hakeem et al., where overexpression of p16INK4a was observed without evidence of HPV being found [38].

With bioinformatic analyses of gene expression databases, large amounts of information can be organized, to determine candidates that can be associated with etiopathogenic phenomena of odontogenic cysts and tumors, or even as candidates for therapeutic targets, which allows us to get one step closer to understanding and treating these entities. However, the principal limitation of this in silico analysis was the hub genes validation by in vitro or in vivo assays. These validations should be performed at genomic, transcriptomic, and/or proteomic levels, considering the resulting candidate gene as a possible diagnosis/prognosis biomarker or therapeutic target. Considering our resulting upregulated and underregulated genes, it is necessary to perform an assay at transcriptomic and proteomic levels to analyze the expression of related genes with the extracellular matrix modification. The clinical potential of the obtained results could impact the prediction of OK recurrence or aggressive growth, and thus postulate more specific treatments that consider biological behavior. Wang et al. showed that SHH inhibitors could be considered as possible pharmacological adjuvants [39]. If we consider that currently obtaining OK cell lines is feasible [40], the study of the downregulated genes related to immune response and viral infection through a three-dimensional or co-culture model, where viral exposure trials can be established, is a new finding that could be validated providing novel information about OK pathogenesis.

## 4. Materials and Methods

The gene expression profile GSE38494 was obtained from the NCBI-GEO database, which is considered a free access and use public database. This profile was obtained from the analysis of sporadic OK (n = 12) and normal gingiva as a control group (n = 2) using the Affymetrix Human Genome U133 Plus 2.0 Array Chip (HG-U133_Plus_2), which contains 54,670 probe sets. The original samples obtained approvals from their respective ethics committees (0/H0703/054 and CPP53-10).

The GEO2R software 4.2.2 was used, applying the Benjamini and Hochberg test to determine the False Discovery Rate (FDR), considering *p* < 0.05 as significant as the cutoff point. Only those genes with a log fold change (FC) >2 and <2 were selected, with a *p* < 0.05.

GO and KEGG Enrichment

Each resulting list of differentially expressed genes (DEG) >2 and <2 was subjected to enrichment analysis on the DAVID 6.8 platform, https://david.ncifcrf.gov (accessed on 17 September 2023), considering the Entrez_Gene_ID as an identifier [41]. Enrichment was performed through gene ontology (GO) analysis of biological processes and canonical signaling pathways were obtained from the Kyoto Encyclopedia of Genes and Genomes (KEGG). For both GO and KEGG, only criteria with a *p* < 0.05 were considered, selecting the first 10 criteria from each analysis.

Protein–protein interaction (PPI)

Both gene lists of >2 and <2 DEG criteria were analyzed to determine protein–protein interactions (PPI), with the STRING tool version 11.0, http://string-db.org/ (accessed on 17 September 2023) [42], used. The interaction listing in TSV format was analyzed in Cytoscape software v 3.8.2 with MCODE application to select the most significant clustering module in IPP networks, considering degree cutoff = 2, node score cutoff = 0.2, k-core = 2, and maximum depth = 100. From the resulting list, only clustered genes were selected, with an MCODE score > 10.

Post-functional enrichment of candidate genes

Each gene list was submitted to the Cytoscape-ClueGO v. 2.5.10 application to obtain the first 10 biological process criteria considering only terms with a *p*-Value corrected with Bonferroni step-down <0.05, and thus obtain the most relevant genes linked to the pathogenesis of OK.

## 5. Conclusions

Considering the results of the enrichment, PPI, and post-enrichment analysis, we can conclude that the upregulated genes (COL5A2, MMP2, COL11A1, COL12A1, COL5A1, COL4A2, COL8A1, COL1A1, COL1A2, COL3A1, and COL6A1) and BST from the downregulated genes are related principally to the modulation of the extracellular matrix and that downregulated genes are associated mainly with the alteration of immune response mechanisms to viral infection (CCL5, TNF, IFIH1, MX1, OAS1, OAS2, RSAD2, and STAT1). The findings related to the extracellular matrix modulation have significant potential as possible therapeutic targets to reduce recurrence, while those related to the immune response and viral infection represent a new biological parameter in OK pathogenesis. It is necessary to validate both findings in an independent population, as well as associate them with clinical variables.

## Figures and Tables

**Figure 1 ijms-25-02379-f001:**
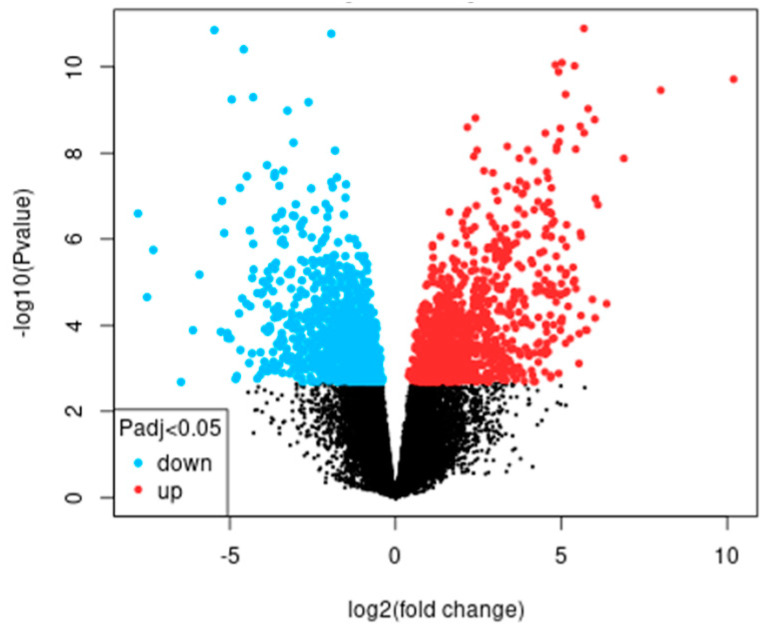
Volcano plot of differentially expressed genes corresponding to greater and less than 2 FC of dataset GSE38494.

**Table 1 ijms-25-02379-t001:** GO biological process terms and KEGG signaling pathways of >2 FC genes.

Category	Terms	Genes Number	*p*-Value	FDR
**GO**	GO:0007155~cell adhesión	69	1.16 × 10^−27^	3.07 × 10^−24^
**GO**	GO:0030199~collagen fibril organization	27	5.47 × 10^−23^	7.23 × 10^−20^
**GO**	GO:0030198~extracellular matrix organization	35	6.05 × 10^−20^	5.34 × 10^−17^
**GO**	GO:0001501~skeletal system development	21	1.46 × 10^−10^	9.67 × 10^−8^
**GO**	GO:0001649~osteoblast differentiation	19	2.87 × 10^−9^	1.52 × 10^−6^
**GO**	GO:0008284~positive regulation of cell proliferation	39	1.14 × 10^−8^	4.83 × 10^−6^
**GO**	GO:0007411~axon guidance	22	1.35 × 10^−8^	4.83 × 10^−6^
**GO**	GO:0001503~ossification	15	1.46 × 10^−8^	4.83 × 10^−6^
**GO**	GO:0071711~basement membrane organization	9	3.42 × 10^−8^	1.01 × 10^−5^
**GO**	GO:0071230~cellular response to amino acid stimulus	12	2.04 × 10^−7^	4.97 × 10^−5^
**KEGG**	hsa04974:Protein digestion and absorption	23	8.90 × 10^−14^	2.18 × 10^−11^
**KEGG**	hsa04512:ECM–receptor interaction	17	3.54 × 10^−9^	4.34 × 10^−7^
**KEGG**	hsa04151:PI3K-Akt signaling pathway	29	9.53 × 10^−7^	7.78 × 10^−5^
**KEGG**	hsa04510:Focal adhesion	21	1.35 × 10^−6^	8.27 × 10^−5^
**KEGG**	hsa05205:Proteoglycans in cancer	18	8.00 × 10^−5^	0.00391793
**KEGG**	hsa04933:AGE-RAGE signaling pathway in diabetic complications	12	1.29 × 10^−4^	0.00527679
**KEGG**	hsa05200:Pathways in cancer	31	2.66 × 10^−4^	0.00932276
**KEGG**	hsa04310:Wnt signaling pathway	15	3.91 × 10^−4^	0.01196999
**KEGG**	hsa05165:Human papillomavirus infection	22	5.28 × 10^−4^	0.01437369
**KEGG**	hsa04360:Axon guidance	15	7.29 × 10^−4^	0.01787028

**Table 2 ijms-25-02379-t002:** GO biological process terms and KEGG signaling pathways of <2 FC genes.

**Category**	**Terms**	**Genes Number**	** *p* ** **-Value**	**FDR**
**GO**	GO:0045087~innate immune response	37	7.88 × 10^−11^	1.64 × 10^−7^
**GO**	GO:0051607~defense response to virus	22	3.48 × 10^−10^	3.62 × 10^−7^
**GO**	GO:0009615~response to virus	15	3.65 × 10^−9^	2.53 × 10^−6^
**GO**	GO:0006954~inflammatory response	26	5.11 × 10^−8^	2.66 × 10^−5^
**GO**	GO:0071346~cellular response to interferon-gamma	12	7.13 × 10^−7^	2.62 × 10^−4^
**GO**	GO:0045071~negative regulation of viral genome replication	9	7.55 × 10^−7^	2.62 × 10^−4^
**GO**	GO:0008544~epidermis development	9	7.99 × 10^−5^	0.02205982
**GO**	GO:0006955~immune response	22	8.48 × 10^−5^	0.02205982
**GO**	GO:0002503~peptide antigen assembly with MHC class II protein complex	5	1.46 × 10^−4^	0.03076427
**GO**	GO:0070106~interleukin−27-mediated signaling pathway	4	1.48 × 10^−4^	0.03076427
**KEGG**	hsa05164:Influenza A	17	7.84 × 10^−7^	1.85 × 10^−4^
**KEGG**	hsa05150:Staphylococcus aureus infection	12	6.37 × 10^−6^	7.52 × 10^−4^
**KEGG**	hsa05330:Allograft rejection	8	1.41 × 10^−5^	0.00110822
**KEGG**	hsa05145:Toxoplasmosis	11	1.46 × 10^−4^	0.00864153
**KEGG**	hsa05332:Graft-versus-host disease	7	2.58 × 10^−4^	0.00994006
**KEGG**	hsa04612:Antigen processing and presentation	9	2.64 × 10^−4^	0.00994006
**KEGG**	hsa04940:Type I diabetes mellitus	7	2.95 × 10^−4^	0.00994006
**KEGG**	hsa05169:Epstein–Barr virus infection	14	4.22 × 10^−4^	0.01244118
**KEGG**	hsa05310:Asthma	6	4.81 × 10^−4^	0.01261624
**KEGG**	hsa05160:Hepatitis C	12	5.77 × 10^−4^	0.01361529

**Table 3 ijms-25-02379-t003:** Group of genes analyzed using the MCODE algorithm.

Category	Genes
Genes derived from >2FC	COL6A3, FBLN1, DCN, THBS2, COL1A1, COL1A2, COL3A1, FN1, POSTN, COL5A1, LUM, COL4A1, COL6A1, COL6A2, NID1, COL5A2, COL14A1, COL15A1, COL8A1, FMOD, COL12A1, NID2, COL11A1, COL4A2, SERPINH1, FBN1, FBLN2, MMP2, CCN2, COL10A1, LAMA4, THBS1, COL16A1, COL18A1, LOXL1, ELN, ADAMTS2, COL21A1, TIMP3, CDH11, THY1, CXCL12, SPP1, PXDN, MMP13, COMP
Genes derived from <2FC	IRF1, CXCL10, OAS2, STAT1, MX1, OAS1, RSAD2, GBP2, GBP1, IFIH1, IFIT3, CCL5, PARP9, DDX60, CXCL9, GBP4, NLRC5, BST2, IFI27, TNF, GBP3, MX2, AIM2, SAMD9
The MCODE algorithm was applied to enrich clusters of the IPP network and identify proteins that are densely connected (MCODE score > 4).	

**Table 4 ijms-25-02379-t004:** Post-enrichment analysis of biological processes using ClueGO of clustered upregulated genes.

ID	Terms	*p*-Value with Bonferroni Correction	Genes Number	Associated Genes
GO:0030198	extracellular matrix organization	5.623 × 10^−42^	31	ADAMTS2, CCN2, COL10A1, COL11A1, COL12A1, COL14A1, COL15A1, COL16A1, COL18A1, COL1A1, COL1A2, COL3A1, COL4A1, COL4A2, COL5A1, COL5A2, COL8A1, COMP, ELN, FBLN1, FBLN2, FMOD, LOXL1, LUM, MMP13, MMP2, NID1, NID2, POSTN, PXDN, SERPINH1
GO:0043062	extracellular structure organization	6.005 × 10^−42^	31	ADAMTS2, CCN2, COL10A1, COL11A1, COL12A1, COL14A1, COL15A1, COL16A1, COL18A1, COL1A1, COL1A2, COL3A1, COL4A1, COL4A2, COL5A1, COL5A2, COL8A1, COMP, ELN, FBLN1, FBLN2, FMOD, LOXL1, LUM, MMP13, MMP2, NID1, NID2, POSTN, PXDN, SERPINH1
GO:0045229	external encapsulating structure organization	7.033 × 10^−42^	31	ADAMTS2, CCN2, COL10A1, COL11A1, COL12A1, COL14A1, COL15A1, COL16A1, COL18A1, COL1A1, COL1A2, COL3A1, COL4A1, COL4A2, COL5A1, COL5A2, COL8A1, COMP, ELN, FBLN1, FBLN2, FMOD, LOXL1, LUM, MMP13, MMP2, NID1, NID2, POSTN, PXDN, SERPINH1
GO:0030199	collagen fibril organization	5.279 × 10^−25^	15	ADAMTS2, COL11A1, COL12A1, COL14A1, COL1A1, COL1A2, COL3A1, COL5A1, COL5A2, COMP, FMOD, LOXL1, LUM, PXDN, SERPINH1
GO:0035987	endodermal cell differentiation	1.849 × 10^−15^	10	COL11A1, COL12A1, COL4A2, COL5A1, COL5A2, COL6A1, COL8A1, FN1, LAMA4, MMP2
GO:0001706	endoderm formation	1.154 × 10^−14^	10	COL11A1, COL12A1, COL4A2, COL5A1, COL5A2, COL6A1, COL8A1, FN1, LAMA4, MMP2
GO:0007492	endoderm development	5.997 × 10^−13^	10	COL11A1, COL12A1, COL4A2, COL5A1, COL5A2, COL6A1, COL8A1, FN1, LAMA4, MMP2
GO:0001704	formation of primary germ layer	2.526 × 10^−11^	10	COL11A1, COL12A1, COL4A2, COL5A1, COL5A2, COL6A1, COL8A1, FN1, LAMA4, MMP2
GO:0071230	cellular response to amino acid stimulus	3.293 × 10^−10^	8	COL16A1, COL1A1, COL1A2, COL3A1, COL4A1, COL5A2, COL6A1, MMP2
GO:0043200	response to amino acid	4.674 × 10^−10^	9	CCN2, COL16A1, COL1A1, COL1A2, COL3A1, COL4A1, COL5A2, COL6A1, MMP2

**Table 5 ijms-25-02379-t005:** Post-enrichment analysis of biological processes using ClueGO of clustered underregulated genes.

ID	Terms	*p*-Value with Bonferroni Correction	Genes Number	Associated Genes
GO:0009615	response to virus	1.985 × 10^−30^	21	AIM2, BST2, CCL5, CXCL10, CXCL9, DDX60, GBP1, GBP3, IFI27, IFIH1, IFIT3, IRF1, MX1, MX2, NLRC5, OAS1, OAS2, PARP9, RSAD2, STAT1, TNF
GO:0051607	defense response to virus	2.420 × 10^−28^	19	AIM2, BST2, CXCL10, CXCL9, DDX60, GBP1, GBP3, IFI27, IFIH1, IFIT3, IRF1, MX1, MX2, NLRC5, OAS1, OAS2, PARP9, RSAD2, STAT1
GO:0140546	defense response to symbiont	2.483 × 10^−28^	19	AIM2, BST2, CXCL10, CXCL9, DDX60, GBP1, GBP3, IFI27, IFIH1, IFIT3, IRF1, MX1, MX2, NLRC5, OAS1, OAS2, PARP9, RSAD2, STAT1
GO:0034341	response to interferon-gamma	4.479 × 10^−16^	11	BST2, CCL5, GBP1, GBP2, GBP3, GBP4, IRF1, NLRC5, PARP9, STAT1, TNF
GO:0071346	cellular response to interferon-gamma	9.175 × 10^−15^	10	CCL5, GBP1, GBP2, GBP3, GBP4, IRF1, NLRC5, PARP9, STAT1, TNF
GO:0071346	cellular response to interferon-gamma	9.175 × 10^−15^	10	CCL5, GBP1, GBP2, GBP3, GBP4, IRF1, NLRC5, PARP9, STAT1, TNF
GO:0048525	negative regulation of viral process	4.501 × 10^−14^	9	BST2, CCL5, IFIH1, MX1, OAS1, OAS2, RSAD2, STAT1, TNF
GO:0045071	negative regulation of viral genome replication	1.100 × 10^−13^	8	BST2, CCL5, IFIH1, MX1, OAS1, OAS2, RSAD2, TNF
GO:0019079	viral genome replication	9.466 × 10^−13^	9	BST2, CCL5, IFI27, IFIH1, MX1, OAS1, OAS2, RSAD2, TNF
GO:0045069	regulation of viral genome replication	3.289 × 10^−12^	8	BST2, CCL5, IFIH1, MX1, OAS1, OAS2, RSAD2, TNF

## Data Availability

The supporting data for reported results can be found in DataSets for GSE38494, from 12 sporadic keratocystic odontogenic tumors, and two control samples were hybridized on Affymetrix^®^ GeneChip Human Genome U 133 Plus 2.0 arrays on line available at: https://www.ncbi.nlm.nih.gov/geo/query/acc.cgi?acc=GSE38494 (accessed on 2 December 2023).

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
