# Peer review of "In Silico Analysis of Genes Associated with the Pathogenesis of Odontogenic Keratocyst"

_ijms, 2024, doi:10.3390/ijms25042379_

Round 1
Reviewer 1 Report
Comments and Suggestions for Authors
The main question that the research addresses is, " Are there specific and identifiable genes in the pathogenesis of odontogenic keratocyst using the gene expression profile from the NCBI-GEO database? "”
The relevance of this manuscript is the gene expression profile of odontogenic keratocyst was consciously analyzed using a robust methodology to identify specific gene expressions and protein interactions. Since there is no previously reported similar analysis, it is a novel approach.
The methodology is accurate and presented clearly and understandably. The conclusions are limited but concise and based on the results presented and discussed. The suggestion of the need for further validation makes them realistic and non-misleading. The references are up-to-date, appropriate, and sufficient, and no unnecessary self-citations were detected.
Tables provide non-repeated information, and the format and extension make them understandable. The figure presents a plot for which this reviewer has no additional comments.
It is a well-written and interesting manuscript.
In line 140, the closing bracket of the in-text reference 13 is missing.
Author Response
Respected reviewers
Thank you very much for your valuable and extremely timely observations. These were well received by all of us since they allowed us to see this manuscript in a better way. We grouped these observations since they were coincident.
- In line 140, the closing bracket of the in-text reference 13 is missing.
- Thanks for the observation, the correction was made.
- Expand details on methods—samples, statistics, ethics
- Additional information was provided (Line 233-235). Regarding the question of ethics, it is important to note that since it is a free and publicly accessible database, the data corresponding to the ethical approval numbers were taken from this same database.
- Enhance presentation and discussion of results
- Additional information was provided in “2.3 Post-functional enrichment of candidate genes” and in discussion section (lines 175-197)
- Summarize conclusions and limitations clearly
- Additional information was provided in lines 214-228 and 261-268.
- Revise language/gramar
- Thanks for the observation, additional revisions were made.
- Introduction is very short, please extend it, you can describe in more details about country/part of the world, sex, age, more affected with OK.
- Additional information in introduction was provided (lines 58-68 and 79-84)
- Please include a small table with the clinical characteristics and etiology of the OK.
- We regret not being able to include this table that you kindly suggest, when constructing it we observed that we redounded with respect to the text included in the introduction and discussion.
- Make another paragraph mentioning future or further studies about your study.
- Particular information was added in lines 217-228.
Once again, we appreciate your comments and remain attentive to any additional changes.
Kindly
Authors
Reviewer 2 Report
Comments and Suggestions for Authors
Dear authors, I have completed my review of your manuscript, "In-silico analysis of genes associated with the pathogenesis of odontogenic keratocyst," submitted for consideration in the International Journal of Molecular Sciences. Please find below my comments after assessing your paper: The study examines an important research question regarding the underlying genetic factors influencing odontogenic keratocyst pathogenesis. The bioinformatics analysis methodology using publicly available gene expression data is sound. However, certain aspects need to be addressed to enhance the clarity, interpretation, and impact of the findings: Introduction Well-written background with a concise overview of current knowledge on odontogenic keratocysts. There are no major recommendations. Methods Describe statistical and bioinformatics analyses in greater depth - analysis flow, software parameters, and validation methods. Comment on ethical approvals obtained for the use of human tissues in line with journal requirements (include the number). Results Expand interpretation of analysis results - agreements or contrasts with literature.
Discussion Relate study findings to previous evidence more deeply. Discuss limitations - small sample size, lack of experimental validation, etc. Conclusion Summarize major findings and significance concisely. Language Revise English language usage throughout the manuscript with a focus on grammar. Overall, this study presents meaningful findings that can advance the understanding of odontogenic keratocyst pathogenesis after addressing the recommendations raised in my review. Please feel free to contact me if you have any questions regarding this review. I look forward to assessing a revised draft of your work.
Key recommendations:
- Expand details on methods—samples, statistics, ethics
- Enhance presentation and discussion of results
- Summarize conclusions and limitations clearly
- Revise language/grammar
Moderate editing of English language required
Author Response

(The authors gave the same response as above.)

Reviewer 3 Report
Comments and Suggestions for Authors
Dear authors, thank you for submitting the manuscript "In-silico analysis of genes associated with the pathogenesis of odontogenic keratocyst".
You manuscript displays high quality of work, however there are a few things that can improve and make it more interesting to the readers. Here is my feedback:
Some editing for English language is required throughout the manuscript.
Introduction is very short, please extend it, you can describe in more details about country/part of the world, sex, age, more affected with OK.
Please include a small table with the clinical characteristics and etiology of the OK.
Make a paragraph mentioning all the limitations of the study.
Make another paragraph mentioning future or further studies about your study.
The article can be more attractive if you include an image or drawing of the common locations in the mouth affected by the OK.
Comments on the Quality of English LanguageMinor editing of English language required
Author Response

(The authors gave the same response as above.)

Round 2
Reviewer 2 Report
Comments and Suggestions for Authors
Dear authors,
I have completed a re-review of your revised manuscript titled "In-silico analysis of genes associated with the pathogenesis of odontogenic keratocyst," submitted to the IJMS. You have addressed several of the concerns raised in my initial review. Please find below my remaining comments:
Major Revisions:
- The English language usage has improved but still contains issues with grammar, sentence structure, etc., in parts. Please thoroughly proofread the manuscript.
- The discussion section relates the current findings to previous literature more deeply, but further expansion is needed in interpreting the specific results of this study.
- The limitations should comment on the lack of experimental validation and functional assays to confirm the pathogenic roles of identified genes.
Minor Revisions:
- There are still some minor formatting inconsistencies in the references. Please double-check journal abbreviations/formatting.
- The conclusion now effectively summarizes the major results, but it would be strengthened by highlighting the potential significance of these findings.
Overall, the study methodology is appropriate, and the manuscript has improved a lot. Addressing the remaining language issues and clearly noting limitations around experimental validation would further enhance this work. Provided these final edits are made, I believe this manuscript could represent a solid contribution worthy of publication after revision.
Comments on the Quality of English LanguageMinor editing of the English language is required.
Author Response
Dear reviewer
We sincerely thank you for your review, as well as apologize for the omissions presented in the previous revision.
We list the modifications made, staying tuned for additional changes.
Major Revisions:
- The English language usage has improved but still contains issues with grammar, sentence structure, etc., in parts. Please thoroughly proofread the manuscript.
- An additional revision was performed.
- The discussion section relates the current findings to previous literature more deeply, but further expansion is needed in interpreting the specific results of this study.
- Additional discussion relates to MMP2 and BST2 (lines 184-186), CCL5 and TNF (line 200-203)
- The limitations should comment on the lack of experimental validation and functional assays to confirm the pathogenic roles of identified genes.
- The limitations paragraph was restructured with new information related possible assay that could be performed (228-233).
Minor Revisions:
- There are still some minor formatting inconsistencies in the references. Please double-check journal abbreviations/formatting.
- The references format was modified, and abbreviation revised.
- The conclusion now effectively summarizes the major results, but it would be strengthened by highlighting the potential significance of these findings.
- The potential significance was added (line 272-275).
We are at your disposal for any comments, recognizing that it was very pleasant to work with you.
Sincerely